# Evaluation of Key Positions and Employees Management Level in Manufacturing Industry—The Czech Case

**Petra Horváthová [1],\*** , **Šárka Velčovská [2]**, **Lenka Kauerová [2] and Friðrik Rafn Larsen [3]**

[1] Department of Management, Faculty of Economics, VSB-Technical University of Ostrava, Sokolská třída 33, 702 00 Ostrava 1, Czech Republic
[2] Department of Marketing and Business, Faculty of Economics, VSB-Technical University of Ostrava, Sokolská třída 33, 702 00 Ostrava 1, Czech Republic; sarka.velcovska@vsb.cz (S.V.); lenka.kauerova@vsb.cz (L.K.)
[3] School of Social Sciences, Faculty of Business Administration, University of Iceland, Sæmundargata 2, 102 Reykjavík, Iceland; fl@hi.is
\* Correspondence: petra.horvathova@vsb.cz

**Abstract:** Human resources management, especially the key employees management, has fundamental influence on companies' sustainable business, which has to be considered as the priority of any business functioning. The aim of this paper is to evaluate current level of the key positions and employees management in the Czech business environment and to propose a process of its effective implementation and application in practice. Online questionnaire survey was provided using 105 companies operating in manufacturing industry of the Moravian-Silesian region as the sample. Statistical methods of data analysis were used. Fisher exact test, coefficients Phi and Cramer's V were counted to test relations between variables. The survey results include an evaluation of the present situation as it comes to the use of the key positions and employees management system as well as an identification of interests in new system implementation. There was a low rate of use of given system in these business, with two thirds of companies showing an interest in implementing the new system. The process of implementation and the use of the key positions and employees in practice was proposed. The process is suggested in the way to make its content and form be a certain guide and help for companies to realize particular activities of this process. It would contribute to the successful realization of complex and systematic approach towards the work with the key positions and employees.

**Keywords:** key position; key employee; process; system; management; manufacturing industry; sustainable business

## 1. Introduction

Sustainable entrepreneurship based on the principles of sustainable development also includes social sustainability, expressed in personnel policy, which includes management and care for all employees, but especially the key ones, as the functioning of the company is based on them. If the company fails to properly manage and maintain them, these employees may leave the company, which may have significant implications in very sensitive areas such as business strategy, achievement of goals, company culture, or the morale of ordinary employees. The company may suffer significant financial loss by the departure of key employees, its economic (and thus environmental) sustainability or even its existence may be jeopardized. On the other hand, well-managed, strategically linked

and well executed management of key employees and positions becomes a significant competitive advantage for successful companies and can contribute to its sustainable business.

Those companies, which understand the significance of sustainable HR management and quality (mainly the key human resources), and companies, which determined themselves to take care of the key positions and employees and to accept it to be one of their basic priorities (having finances to do so), should use a complex and systematic approach [1]. Such approach directly includes and connects mainly processes of identification of the key positions, the key competences, acquiring the key employees out of internal and external sources, employees qualification development and consequent sustaining in company [2].

It is necessary to take into account that key positions and employees management do not act alone and separately in the company; they are always very closely connected with the management of talented employees (talent management). One of the main contributions of talent management, more effective planning of the key positions is as well as gaining the employees from internal sources including their motivation. The result of well-functioning talent management, which has to be based on the model Develop-Deploy-Connect (DDC) is then the existence of sufficient amount of candidates—key employees—prepared to work at the key positions. Contrary to traditional linear model of talent management, the model DDC does not focus only on gaining and maintaining but mainly on developing employees' skills, their deploying to such positions where those will be able to use their potential and knowledge and their connection with people who can help them with task fulfillment, too [2,3]. Quality and sustainable management of the key positions and employees is, thus, conditioned by quality processes of talent management (acquisition, development and retention).

To find out the real situation in the area of management of key positions and employees, the authors decided to carry out a research study, which will evaluate the attitudes of manufacturing companies in one of the regions of the Czech Republic to this issue and also propose a process of effective implementation and application of key positions and employees management in practice. To the best of authors' knowledge, there are no research studies, which would deal with the rate of the usage of the key positions and employees management system in companies, neither in the Czech Republic, nor abroad.

The structure of the article is as follows. The literature review provided first followed by research methodology explanation; further, there are results presented and proposal for the process of implementation and application of the key positions and employees system is given.

## 2. Literature Review

Key position is to be determined by four criteria. Naturalness—short-time failure of employee working at this position can bring a serious impact on economic or business indicators, e.g., yields, profit, loss of competitive advantage, limitation of operation, putting process, products or services in danger, etc. Uniqueness—position demands such competences being now or in the future unique and specific for the company. Demand—for such position, there is or will be a huge demand in the labor market in the future. Strategic impact—failure of qualified employees working at concrete positions would influence the company success in the future considering a longer time horizon [4].

According Skills-Based Workforce Segmentation Model based on the most prominent conceptual HRM architectural model [5,6] defines key positions as those having higher skills value and higher skills uniqueness. Valuable skills may create lower costs, increase revenue, contribute to innovation, or enhance internal company efficiency. They comprise up to nine key value drivers—revenue/sales, stakeholder relations, cost and efficiency, quality, innovation, organizational capability, reputation/risk management, financial, and processes/systems. Various roles will comprise one or more of these value drivers, with their impact ranging from localized to widespread across the organization. Some of these value drivers may be more important than others to the achievement of the business strategy at a particular point in time. Unique skills are organizational specific, unlikely to be found in the open market, hard to replace, and may be difficult for competitors to imitate or duplicate. These skills

need to be nurtured over time, given that they are not developed and acquired overnight. Hence organizations are more likely to invest in the education, training, and development of these skills [6].

Companies should pay eminent attention to the key positions because an individual performance of those positions has deciding impact on the entire company performance and sustainable business [7,8]. The fundamental premise to define the key positions is the fact that the company cannot fully function without it [9,10]. Those positions are considered as core of the future company success adding that there is a significant need to pay attention to gaining, evaluation, development and sustaining of the individuals of those positions, thus the key employees through entire company [11]. Without the key employees the operation of the company is put in danger in significant way because without those employees some activities cannot even be provided [12]. The key employees play a significant role as the best performing part of HR, have quality professional knowledge and skills, which they can share with other colleagues of less competences [13,14]. Such employees are special also because of the high level of self-control and self-reflection. The quality of the key employees influences not only entire company performance and sustainable business but its good reputation, brand and company capital's value as well [15]. Once those leave, unique know how is disappearing from the company as well as their knowledge, abilities, skills experiences and long-term relationships [16,17]. Therefore, each company should manage those positions and employees in a way not only to keep the current key employees but to train own successors for those critical and strategic positions as well [10,18,19]. The significance of the key positions and employees is supported by the statement that the fastest way to make shareholders' wealth grow and improvement of long-term company's sustainable business is to have the key employees' performance became higher [20,21].

Significance of the key positions and employees is confirmed not only by literature review mentioned above, but by many studies and researches as well. For example, the study North American Critical Talent Practices Survey says that definition and assigning the critical positions will play more significant role in the future, as it comes to company competitiveness, than it has been in the past and at present time [22]. By the study Rewriting the Rules for the Digital Age. 2017 Deloitte Global Human Capital Trends, companies currently facing the changing context of labor force and labor world should pay big attention to the human capital strategies, thus to how they organize, manage, develop and maintain their employees, mainly those talented and key ones [23]. As it comes to the Czech business environment, e.g., the research Czech CEO Survey 2018 dealt with this topic. It considers the qualified key employees management the strategic task, which should be paid attention to by the management of each company [24]. In the Czech business environment, systematic approach towards the key positions and employees management is not being applied sufficiently [25].

According to the authors opinion, one of the reasons why such approach shows rare application in the Czech business environment can be the absence of instructions how this process should be implemented and applied. Therefore, the authors decided to prove this conclusion by the research, the aim of which is mapping the current approaches to the key positions employees management in middle and big companies of Moravian-Silesian region operating in the manufacturing industry. Then the identification of companies' interest in the key positions and employees management system served as a stimulus for proposing the procedure of implementation and application of this system in practice.

## 3. Sample and Methodology

The survey was conducted to find out whether and in what extent small and medium-sized companies operating in manufacturing industry of the Moravian-Silesian region manage the key positions and employees. Its objective was to determine the state, way and length of the use of the key positions and employees management system in companies, to evaluate satisfaction with this system, to identify reasons for non-use of the system, and to find out interest of companies in implementing new management system for the key positions and employees.

Primary data were collected in April 2018 using an online questionnaire survey based on a structured questionnaire. In the introductory part of the questionnaire the respondents were explained what the system of managing key positions and employees means, what is the principle of the system and what are its benefits. The questionnaire included three identification questions (length of an company existence, current number of employees, foreign owner/shareholder) and six questions on the subject of the research, which were closed (system usage/non-use, length of the system usage, evaluation of success and benefits of system used, interest in implementation of a new system) or semi-closed with the possibility of writing own response. Semi-closed questions were used to find out the current way of managing key positions and employees in the company and to identify the reasons for non-using the system.

The target group for the research consisted of all medium-sized (51–250 employees) and large companies (more than 250 employees) operating according to the CZ-NACE classification in manufacturing industry in the Moravian-Silesian region. As of 31 December 2017, it was a total of 366 companies, of which 259 (i.e., 70.8%) were medium-sized and 107 (i.e., 29.2%) were large companies [26]. The sample included 105 companies (i.e., 28.7% of the target group) selected by the quota sampling technique with respect to the above-mentioned size structure of companies. The final sample structure is presented in Table 1. Competent representatives of the personnel departments of the companies were interviewed.

**Table 1.** Structure of respondents.

| Variable | | Absolute Frequency | Relative Frequency |
|---|---|---|---|
| Length of an company existence | 2 to 5 years | 20 | 19.0% |
| | 6 to 10 years | 27 | 25.7% |
| | 11 to 15 years | 36 | 34.3% |
| | more than 15 years | 22 | 21.0% |
| Current number of employees | 51 to 250 employees | 75 | 71.4% |
| | more than 250 employees | 30 | 28.6% |
| Foreign owner/shareholder | yes | 23 | 21.9% |
| | no | 82 | 78.1% |

Data were processed and analyzed in IBM SPSS Statistics 23.0 program. For selected questions, the second level grading according to the respondent's identification characteristics or relevant variables was performed. Fisher exact test at 0.05 significance level was applied to test the differences between variables. This test is suitable for nominal variables in the analysis when sample size is small [27]. The hypothesis H0 presumes no statistically significant differences among variables; the hypothesis H1 presumes dependency of variables. Cramer's V coefficient and Phi coefficient were counted to prove the strength of relationship between variables. The Cramer's V coefficient is used for nominal variables when the number of rows or columns or both in the contingency table is higher than 2. The Phi coefficient ($\varphi$) is a measure of association for two binary variables, i.e., for a $2 \times 2$ contingency table. These descriptors were used to interpret the coefficients: 0.00 and under 0.09—negligible association, 0.10 and under 0.19—weak association, 0.20 and under 0.39—moderate association, 0.40 and under 0.59—relatively strong association, 0.60 and under 0.79—strong association, 0.80 and under 1.00—very strong association [28].

## 4. Results and Discussion

In accordance with the objective of research study, the results are structured into three following areas: (1) The current state of use of the key positions and employees management system in companies, (2) satisfaction with the existing system of managing the key positions and employees and (3) interest in implementation of new management system for the key positions and employees.

### 4.1. The Current State of Use of the Key Positions and Employees Management System in Vompanies

In this part of the research study, respondents were asked whether, how long and in what way they are currently using the management system for the key positions and employees. The management system for the key positions and employees have implemented only 21.0% (i.e., 22) of respondents from the 105 companies addressed. The system is used by 37.0% (i.e., 10) of companies operating on the market for 6 to 10 years and by a quarter of companies (i.e., 5) operating for 2 to 5 years. Almost 90% (i.e., 51) of companies existing 11 or more years do not use such management system. From the point of view of property structure of the companies, it was found that the system is applied in 34.8% (i.e., 8) of companies with a foreign owner or shareholder and only 17.1% (i.e., 14) of Czech companies. Further, huge companies over 250 employees use the system more often (30.0%, i.e., 9) than medium-sized companies with 51 to 250 employees (17.3%, i.e., 13). It is obvious that the management system of the key positions and employees is more common in huge companies, in companies with a foreign owner or shareholder and operating on the market for less than 10 years (Table 2).

**Table 2.** Use of the management system for the key positions and employees by type of company.

| Characteristics of Company | | Use of System | | | |
|---|---|---|---|---|---|
| | | Absolute Frequency | | Relative Frequency | |
| | | Yes | No | Yes | No |
| Length of an company existence | 2 to 5 years | 5 | 15 | 25.0% | 75.0% |
| | 6 to 10 years | 10 | 17 | 37.0% | 63.0% |
| | 11 to 15 years | 4 | 32 | 11.1% | 88.9% |
| | more than 15 years | 3 | 19 | 13.6% | 86.4% |
| Current number of employees | 51 to 250 employees | 13 | 62 | 17.3% | 82.7% |
| | more than 250 employees | 9 | 21 | 30.0% | 70.0% |
| Foreign owner/shareholder | yes | 8 | 15 | 34.8% | 65.2% |
| | no | 14 | 68 | 17.1% | 82.9% |

It was confirmed by Fischer exact test that there is no statistically significant relation between the use of the system and length of the company existence, property structure of company or actual number of employees, Exact Sig. > 0.05. Therefore, the hypothesis H0 is accepted.

Respondents who have the management system for the key positions and employees were asked about length and way of the system use. Approximately half of companies (54.5%, i.e.,12) respondents work with the system for less than 2 years, 45.5% (i.e., 10) respondents have the system for 2 to 5 years. It was also found that 54.5% (i.e., 12) of companies apply the system based on the established procedure and 45.5% proceed non-systematically, with respect to their current needs. The dependence of way of managing the key positions and employees on length of the system use was confirmed (Exact Sig. = 0.000), the value of the Phi coefficient (Phi = 0.833) indicates a very strong association between the variables. Therefore, we reject the hypothesis H0 and accept the alternative hypothesis H1. Companies using the system for a short period of time (up to 2 years) apply a non-systematic approach (83.3%, i.e., 10), while companies working with the system for 2 to 5 years have the established procedure to manage the key positions and employees (100.0%, i.e., 10 of respondents) (Table 3).

**Table 3.** Way of managing the key positions and employees by length of the system use.

| Way of Managing the Key Positions and Employees | Length of the System Use | | | |
| --- | --- | --- | --- | --- |
| | Absolute Frequency | | Relative Frequency | |
| | Less Than 2 Years | 2 to 5 Years | Less Than 2 Years | 2 to 5 Years |
| based on the established procedure | 2 | 10 | 16.7% | 100.0% |
| non-systematic, according to actual needs | 10 | 0 | 83.3% | 0.0% |

Way of managing the key positions and employees does not depend on length of the company existence, on size of the company (i.e., current number of employees) or on property structure of the company, Exact Sig. > 0.05. Therefore, the hypothesis H0 is accepted.

Respondents without the system of managing the key positions and employees reported the reasons for non-use of the system. Two thirds (68.7%, i.e., 57) of them do not know the system, for 15.7% (i.e., 13) the system is organizationally too demanding, 8.4% (i.e., 7) companies do not consider the system suitable for their business and 7.0% (i.e., 6) of respondents see the problem in lack of personnel resources.

*4.2. Satisfaction with the Existing System of Managing the Key Positions and Employeesthe Current State of Use of the Key Positions and Employees Management System in Vompanies*

Respondents using the key positions and employees management system were also asked how they are satisfied with the system. Exactly half of them evaluate the system positively and consider them almost or fully successful and contributive for their company, while the remaining 50.0% (i.e., 11) perceive the system almost or fully not successful and not contributive.

Statistical analysis using Fischer exact test has shown that there is no dependence of the system's evaluation on size of the company, on length of the company existence or on property structure of the company, Exact Sig. > 0.05. Therefore, we accept the hypothesis H0. On the contrary, a statistically significant relationship between the evaluation of the system and length of its use was confirmed (Exact Sig. = 0.000), Cramer's V value = 0.937 shows a very strong association between the variables. There is also a relationship between the evaluation of the system and existing way of managing the key positions and employees (Exact Sig. = 0.002), according to Cramer's V = 0.776, a strong association between variables is proven. In these two cases, therefore, we reject the hypothesis H0 and accept the alternative hypothesis H1. The longer use of the key positions and employees management system by respondents, the more positive evaluation of the system. Further, those respondents who manage the key positions and employees using the established procedure evaluate the system positively, 83.4% (i.e., 10) of them consider the system almost or fully successful and contributive. Conversely, 90.0% (i.e., 9) of companies managing the key positions and employees non-systematically assess the system almost or fully not successful and not contributive (Table 4).



**Table 4.** Evaluation of current system by length of the system use and way of managing the key positions and employees.

| Evaluation of Current System | Length of the System Use | | | | Way of Managing the Key Positions and Employees | | | |
|---|---|---|---|---|---|---|---|---|
| | Absolute Frequency | | Relative Frequency | | Absolute Frequency | | Relative Frequency | |
| | Less Than 2 Years | 2 to 5 Years | Less Than 2 Years | 2 to 5 Years | Based on the Established Procedure | Non-Systematic, According to Actual Needs | Based on the Established Procedure | Non-Systematic, According to Actual Needs |
| fully not successful and fully not contributive | 3 | 0 | 25.0% | 0.0% | 0 | 3 | 0.0% | 30.0% |
| almost not successful and almost not contributive | 8 | 0 | 66.7% | 0.0% | 2 | 6 | 16.7% | 60.0% |
| almost successful and almost contributive | 1 | 2 | 8.3% | 20.0% | 2 | 1 | 16.7% | 10.0% |
| fully successful and fully contributive | 0 | 8 | 0.0% | 80.0% | 8 | 0 | 66.7% | 0.0% |

*4.3. Interest in Implementation of New Management System for Key Positions and Employees*

The key question of the survey, which was answered by all 105 respondents, was focused on identifying respondents' interest in implementation of the key positions and employees management system that would minimize the barriers for its use or would be more efficient in comparison with the current practice applied in the company. Respondents' reactions were positive; most of them (65.7%, i.e., 69) are interested in such system. Interest in implementing new management system for the key positions and employees does not depend on size of the company, length of the company existence, property structure of the company, or the fact whether the system is currently used by companies, Exact Sig. > 0.05, therefore we accept the hypothesis H0. However, respondents without previous experience with such system (68.7%, i.e., 57) have shown a slightly higher interest in implementation of new management system, compared with respondents who currently manage the key positions and employees (54.5%, i.e., 12).

Interest in implementation of new management system depends on length of the system use (Exact Sig. = 0.000, Phi = 1.000), on way of managing the key positions and employees (Exact Sig. = 0.000, Phi = 0.833), on evaluation of current system (Exact Sig. = 0.000, Cramer's V = 0.937), and on reason for non-use of the system (Exact Sig. = 0.000, Cramer's V = 0.450). In these cases, therefore, we reject the hypothesis H0 and accept the hypothesis H1. According to Phi and Cramer's V coefficient values, a very strong to relatively strong association between the variables were indicated.

Companies using the system less than 2 years (they are the companies managing the key positions and employees rather non-systematically and considering their current system almost or fully not successful and not contributive), have shown a clear interest in implementation of new system (100.0%, i.e., 12 of respondents). Companies that have so far managed the key positions and employees non-systematically have also confirmed interest in new system (100.0%, i.e., 10 of respondents). In addition, new system is attractive for companies that consider their current system almost or fully not successful (100.0%, i.e., 11). If we focus on companies with an established procedure of managing the key positions and employees, only 16.7% (i.e., 2) of them are interested in implementation of new system. Analysis of the reasons for non-use of the system at present has shown that new system is particularly interested in companies that have not yet been aware of such system (82.5%, i.e., 47). Among companies that consider the system too demanding, 46.2% (i.e., 6) of them are interested in new system. One third (i.e., 2) of companies mentioned as the main barrier a lack of resources (especially personnel) and 28.6% (i.e., 2) of companies that have not yet considered the system suitable for their business were interested in new system (Tables 5–7).

**Table 5.** Interest in implementing new system by length of the system use and way of managing the key positions and employees.

| Interest in Implementing New System | Length of the System Use | | | | Way of Managing the Key Positions and Employees | | | |
|---|---|---|---|---|---|---|---|---|
| | Absolute Frequency | | Relative Frequency | | Absolute Frequency | | Relative Frequency | |
| | Less Than 2 Years | 2 to 5 Years | Less Than 2 Years | 2 to 5 Years | Based on the Established Procedure | Non-Systematic, According to Actual Needs | Based on the Established Procedure | Non-Systematic, According to Actual Needs |
| yes | 12 | 0 | 100.0% | 0.0% | 2 | 10 | 16.7% | 100.0% |
| no | 0 | 10 | 0.0% | 100.0% | 10 | 0 | 83.3% | 0.0% |

**Table 6.** Interest in implementing new system by evaluation of current system.

| Interest in Implementing New System | Evaluation of Current System | | | | | | | |
|---|---|---|---|---|---|---|---|---|
| | Absolute Frequency | | | | Relative Frequency | | | |
| | Fully Not Successful and Fully Not Contributive | Almost Not Successful and Almost Not Contributive | Almost Successful and Almost Contributive | Fully Successful and Fully Contributive | Fully Not Successful and Fully Not Contributive | Almost Not Successful and Almost Not Contributive | Almost Successful and Almost Contributive | Fully Successful and Fully Contributive |
| yes | 3 | 8 | 1 | 0 | 100.0% | 100.0% | 33.3% | 0.0% |
| no | 0 | 0 | 2 | 8 | 0.0% | 0.0% | 66.7% | 100.0% |

**Table 7.** Interest in implementing new system by reason for non-use of the system.

| Interest in Implementing New System | Reason for Non-Use the System | | | | | | | |
|---|---|---|---|---|---|---|---|---|
| | Absolute Frequency | | | | Relative Frequency | | | |
| | We Do Not Know the System | The System Is Too Demanding | Lack of Resources, Especially Personnel | The System Is Not Suitable for Us | We Do Not Know the System | The System Is Too Demanding | Lack of Resources, Especially Personnel | The System Is Not Suitable for Us |
| yes | 47 | 6 | 2 | 2 | 82.5% | 46.2% | 33.3% | 28.6% |
| no | 10 | 7 | 4 | 5 | 17.5% | 53.8% | 66.7% | 71.4% |

As follows from the survey results, the rate of utilization of the management system for the key positions and employees in medium and huge companies of the Moravian-Silesian region, operating according to the CZ-NACE classification in the manufacturing industry, is low. Only 21.0% of companies addressed apply the system.

Only a small number of respondents evaluated their experience with the system (22), but it is more important to consider that a large number of companies have no experience with such a system. Therefore, there is the potential to implement it and streamline the processes of managing key positions and employees in these companies. Of the 22 companies that have experience with the system, nearly half of them (45.5%, i.e., 10) manage their key positions and employees in a non-systematic way and exactly half of companies perceive their existing system almost or fully not successful and almost or fully not contributive. The key finding of the survey is interest of companies (65.7%, i.e., 69) in implementing new management system for the key positions and employees, which would minimize existing barriers for its use or would make the existing system applied in the company more efficient.

Interest in new system has been mainly shown by those companies whose experience with their current system is rather short-term, based on a non-systematic approach and rather negative. The companies that have not yet managed the key positions and employees due to lack of knowledge of the system, perceived high organizational difficulty or lack of resources (especially personnel) also demonstrated interest in new system. These companies do not consider mentioned barriers as unrecoverable and believe that an effective management system for the key positions and employees would be able to eliminate them.

Interest in implementation of new management system for the key positions and employees was expressed by medium and huge companies, by Czech companies as well as by companies with a foreign owner or a shareholder, by companies that are relatively young as well as those that have been operating in the market for a long time. Statistical dependence of interest in new system on the type of company was not confirmed. Thus, it can be stated that the key positions and employees management system could be applied across sectors in different types of businesses in small and medium-sized companies operating in manufacturing industry of the Moravian-Silesian region.

In order to obtain more detailed information beyond the outputs of the research study, the results of the research study were subsequently discussed in two focus groups with a total of thirteen HR and TOP business managers (medium and large) participating in the study. The discussion confirmed the conclusions of the research study, i.e., that the management system of key positions and employees can be found more in large companies, in companies with foreign owners or shareholders and operating on the market for less than ten years. The reason for this situation is, according to the focus group participants, mainly because companies do not know the system, they do not know how to work with it. Representatives of those companies that use the system have described in detail the benefits of using the system for their businesses and thus, further boosted the interest in implementing the key positions and employee management system that would minimize the barriers for its use, or would be more efficient in comparison with the current practice applied in the company. Representatives of those companies that use the system but mentioned the main shortcomings in their systems—not a high-quality and detailed strategy of key positions and employee management, which negatively affects the functioning of the whole system, sometimes inaccurate performance and potential evaluation of employees in the process of acquisition of talents, which is reflected in the next talent management process, namely in development of talents, where the proposed talent development activities do not correspond to the actual educational gap of employees involved in the talent management system, or key positions and employee management. As a result, those responsible for the system do not have feedback and cannot initiate improvements to the system or remove partial deficiencies. According to the focus group participants, not only these most important but also other shortcomings of the system could be remedied using a well-elaborated process of key positions and employees management system. This is connected with the most important output of the focus groups, i.e., their participants showed a strong interest in a kind of "manual"—a tool that would be help to implement a system of

managing key roles and employees in their company and detailing the individual steps of the process. For this reason, the authors of the article decided to offer this prepared procedure to interested parties not only from the companies that participated in the research study, but basically to any manufacturing industry company in the Moravian-Silesian Region (see Section 5. Practical Implications).

## 5. Practical Implications

As it was already mentioned above, based on the requirement of the business representatives who participated in the research study and focus groups, the authors propose a procedure for implementing and using this system in practice.

The procedure was created based on both theoretical knowledge and practical experience of the authors and could help the companies to remove certain obstacles preventing the companies from wider application of the key positions and employees management system. By removing the barriers, the company performance and sustainable business could become improved and the stability of companies stronger.

In order to facilitate the understanding of the entire management system of key positions and employees by representatives of those companies that will apply the system and then use it, the mind map of the whole system was prepared first.

It enables to see this topic complexly, to get know the system's structure, to gain an idea of not only partial areas and individual activities of this systematic approach but of their realization, complexity and interconnection as well. The mental map is represented by the summarizing graphic presentation of all partial areas (for example *Basic concepts of key positions and employees management*), sub-areas (for example *Acquisition*) and all individual activities (for example Budget) of the key positions and employees management process (See Figure A1).

As mentioned above, the proposal for the process of the key positions and employees management system is based on mental map; particular steps of this process copy a logical sequence of partial areas, sub-areas and individual activities. The process (Table 8) is suggested in the way to make its content and form a certain guide and help for company, thus, the company would be able to realize particular activities of this process. Each area, sub-area and activity should be specified in detail namely by the description of given area and individual activities including a determination of the aim of those activities, their contribution, realization, participating roles and needed bases. Determination of needed sources, not only financial ones, is the example of one of activities. In this context, the question has to be answered of what amount and how structured the budget of the key positions and employees management system would be. Detailed description of this activity is presented in the Table 9 (just for illustration, only a description of one activity is presented; a detailed description of all areas, sub-areas and activities is available from the authors of the paper).

**Table 8.** Process of implementation and use of the key positions and employees management system.

| No | Area | Sub-Area | Activities |
|---|---|---|---|
| 1 | Basic concepts of key positions and employees management | | Key position<br>Key employee |
| 2 | Strategy of key positions and employees management | | Time horizon<br>Business environment<br>Key positions and employees requirements<br>Budget<br>Activity<br>Implementation plan<br>Management roles |
| 3 | Conditions of key positions and employees management | | Initiation and support (top management)<br>Connection with business strategy<br>Allocation of resources |
| 4 | Processes of talent management | Acquisition | Need and resources of talents<br>Key skills (value and uniqueness)<br>Performance<br>Potential<br>Talent-pool |
| | | Development | Program<br>Realization<br>Evaluation<br>Career management |
| | | Retention | Employer attractiveness<br>Realization of activities<br>Reputation of the company |
| 5 | Key positions and employees management | | Problems of key positions and employees management |
| 6 | Evaluation of key positions and employees management | | Audit<br>Evaluation of success |

**Table 9.** Specification of activity "Allocation of resources".

| Objective of Activity | Resources for Key Positions and Employees Providing. |
|---|---|
| Benefit of activity | Specification of resources needed for the application of key positions and employees management. Determination of the amount of structure of the budget for the application of key positions and employees management. |
| Realization of activity | Introductory (preparatory) stage of key positions and employees management application in the company. |
| Participating roles<br>Materials needed | Top management of the company. Financial director.<br>Business strategy. Key positions and employees strategy. Budgets. |

It is evident that the application of proposed process depends on the specific conditions of the company, how concretely the company approaches the implementation of the use of the key positions and employees management system, whether it would or would not use all ideas and activities of the process or whether it would realize any other partial steps.

## 6. Conclusions

Faster and more frequent changes of external environment place higher and higher demands on companies. For all types of companies, people become the main asset to manage those changes. Without qualified labor force, without employees possessing of needed skills and knowledge, many companies would not only keep up with the competition and sustainable business but they even would not be able to survive.

The key positions and employees management can be considered the one of the fundamental tools of HR management because those positions and employees are deciding factor in case the company wants to keep its performance, sustainable business and stability [7–12,16].

The aim of this paper was to evaluate the current level of the key positions and employees management in companies of manufacturing industry in the Czech Republic, identify their interest in

implementation of the key positions and employees management system as well as to propose effective implementation and application of this system in practice.

The results of realized questionnaire survey showed a low rate of the use of the key positions and employees management system in the Czech companies. Almost a half of those companies manage the key positions and employees non-systematically and consider their present system almost or fully not successful and less or fully not contributive. Two thirds of companies expressed their interest in implementation of new system of the key positions and employees management, which would remove the barriers and make current processes more effective. Based on the analysis of the survey results, it can be said that this system can be applied across medium-sized and large enterprises of the manufacturing industry in the Moravian-Silesian Region. Education is necessary for companies that do not yet know the key roles and employees management system, but this issue is beyond the scope of the article. The output of this paper not only results of the survey is but also proposed process of the key positions and employees management, application of which can help companies to survive and grow, overtake the competition and significantly increase their added value and sustainable business. It is the complex system, which can help to gain and maintain the key employees, build on their strong aspects, reward their successes, provide them with opportunities to develop themselves and make their total effectiveness higher and thus, effectiveness of entire company.

The limitations of the research study lie in following facts. The study was provided in the Czech Republic focused on manufacturing industry only.

In future research, it would be useful to extend the research study also to other industries as well as to execute the research in other regions. It would also be interesting to repeat the study after certain time period to find out whether rate implementation of the key positions and employees management is increasing. Another possible field of research would be an investigation of perceived contributions when using this system for longer period of time. The proposed research activities require the creation and use of a unified methodology of data collection and their analysis in order to compare outputs across sectors, regions and over time.

**Author Contributions:** Conceptualization, P.H.; data curation, formal analysis, P.H. and Š.V.; methodology and resources, Š.V. and F.R.L.; validation, Š.V., L.K. and F.L.L.; formal analysis, F.R.L.; investigation, Š.V. and L.K.; visualization, L.K.; supervision, writing—original draft preparation, writing—review and editing, P.H. All authors have read and agreed to the published version of the manuscript.

**Funding:** This research received no external funding.

**Acknowledgments:** The paper was supported within the Operational Program Education for Competitiveness— Project No. CZ.1.07/2.3.00/20.0296 and the project of the Student Grant Competition at the Economic Faculty of VŠB-Technical University Ostrava SP2019/7.

**Conflicts of Interest:** The authors declare no conflict of interest. The funders had no role in the design of the study; in the collection, analyses, or interpretation of data; in the writing of the manuscript, or in the decision to publish the results.

## Appendix A

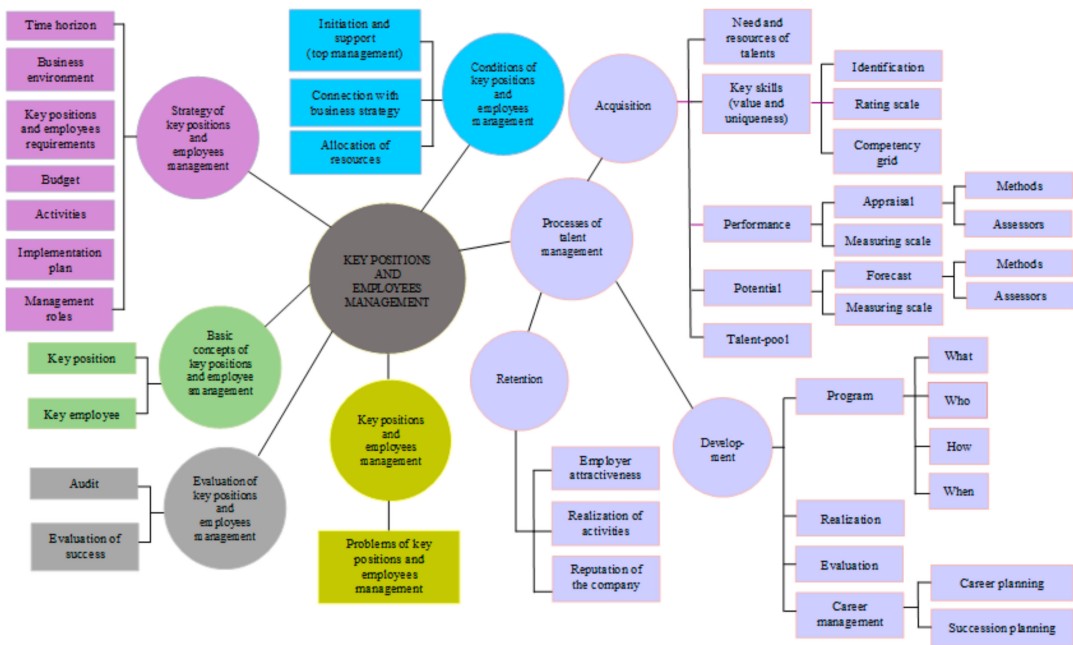

**Figure A1.** Key positions and employees management system mind map.

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
