# Peer review of "Evaluation of Key Positions and Employees Management Level in Manufacturing Industry—The Czech Case"

_sustainability, doi:10.3390/su12010242_

Round 1

Reviewer 1 Report

The paper is accepted.

Author Response

Dear reviewer,

thank you very much for the review.

Authors

Reviewer 2 Report

Dear authors,

It was very interesting to read your manuscript. I think that your work has a potential for publishing. I have several remarks in order to make your scientific article more precise.

The aim of the article was set as an evaluation of the current level of key positions and system of employee management in the Czech business environment (Moravian- Silesian region) and a proposal of the process of its effective implementation and application in practice. An online questionnaire was chosen as the basic tool for data collection. The evaluation of the collected data was processed by the statistical software IBM SPSS and the authors devoted in detail to the statistical evaluation of the obtained data.

The results of the research study include, for example, whether companies use a management system for key positions and employees by enterprise type, how to manage key positions (systematic and non-systematic way), the link between system usage and how they manage key positions, and interest in its implementation.

In the section Practical Implications, the authors propose a procedure for implementation system of management of key positions and employees and write that it is based on the mind map presented in the annex. They further state that the application of the proposed process depends on the specific conditions in the company.

Remarks:

- I did not find a clear link between the research study described in the manuscript and the proposal of implementation process (e.g. how mentioned research study influenced the proposed implementation process and what parameters the new system should have to better meet requirements...)

-In manuscript I did not find a clear link between the results of the research study and the mind map (attachment 1).

- Given that the article contains a proposal for the implementation of the system, I would expect in survey some information on the shortcomings in explored systems.

- I suppose that the authors have conducted a number of interviews not only with HR managers, as they write, but also with TOP managers of selected companies and therefore they could give more detailed results of an evaluation of these interviews, indicating how this information influence functionality of system of key positions and staff management and   proposal of system implementation.                

Conclusions:

I suggest completing the manuscript according to the above remarks.

Author Response

Dear reviewer,

thank you very much for the review and challenging remarks. Furthermore, we commented on individual comments in summary, because the content of the solution of these comments’ overlaps. Based on your recommendations, we added the following paragraph at the end of Chapter 4. Results and Discussion:

In order to obtain more detailed information beyond the outputs of the research study, the results of the research study were subsequently discussed in two focus groups with a total of thirteen HR and TOP business managers (medium and large) participating in the study. The discussion confirmed the conclusions of the research study, i.e. that the management system of key positions and employees can be found more in large companies, in companies with foreign owners or shareholders and operating on the market for less than ten years. The reason for this situation is, according to the focus group participants, mainly because companies do not know the system, they do not know how to work with it. Representatives of those companies that use the system have described in detail the benefits of using the system for their businesses and thus, further boosted the interest in implementing the key positions and employee management system that would minimize the barriers for its use, or would be more efficient in comparison with the current practice applied in the company. Representatives of those companies that use the system but mentioned the main shortcomings in their systems - not a high-quality and detailed strategy of key positions and employee management, which negatively affects the functioning of the whole system, sometimes inaccurate performance and potential evaluation of employees in the process of acquisition of talents, which is reflected in the next talent management process, namely in development of talents, where the proposed talent development activities do not correspond to the actual educational gap of employees involved in the talent management system, or key positions and employee management. As a result, those responsible for the system do not have feedback and cannot initiate improvements to the system or remove partial deficiencies. According to the focus group participants, not only these most important but also other shortcomings of the system could be remedied using a well-elaborated process of key positions and employees management system. This is connected with the most important output of the focus groups, i.e. their participants showed a strong interest in a kind of "manual" - a tool that would be help to implement a system of managing key roles and employees in their company and detailing the individual steps of the process. For this reason, the authors of the article decided to offer this prepared procedure to interested parties not only from the companies that participated in the research study, but basically to any manufacturing industry company in the Moravian-Silesian Region (see chapter 5. Practical Implications).

At the beginning of Chapter 5. Practical Implications we have replaced the first sentence by the following one:

As it was already mentioned above, based on the requirement of the business representatives who participated in the research study and focus groups, the authors propose a procedure for implementing and using this system in practice.

At the beginning of second paragraph of Chapter 5. Practical Implications we have replaced the first sentence by the following one:

In order to facilitate the understanding of the entire management system of key positions and employees by representatives of those companies that will apply the system and then use it, the mind map of the whole system was prepared first.

Authors

Reviewer 3 Report

Dear Authors,

The paper is clearly and completely structured. The analysis of the literature is exhaustive and up to date, the objectives of the research and the method are well defined.
The description of the results is also very detailed and linked to the conclusions. However, it is difficult to grasp the link between this study and sustainability. The concept is referred to in the text a few times (about 8), but in a generic way so it is impossible to understand the relationship between key positions and employees management and sustainability.
Moreover, which pillar of sustainability do you refer to: environmental, economic or social?
To be consistent with the aims of the Journal, the theme of sustainability must be central or in any case strongly interdependent with the object of the research. I suggest that the authors go into this report before publishing their work.

Author Response

Dear reviewer,

thank you very much for the review and your comments. Based on your recommendations we have replaced the first sentence in the abstract by the following modified sentence:

Human resources management, especially the key employees management, has fundamental influence on companies' sustainable business, which has to be considered as the priority of any business functioning.

We have removed the “company sustainability” from the key words and we added “sustainable business”.

Also, the first paragraph of Introduction part was replaced by the following text:

Sustainable entrepreneurship based on the principles of sustainable development also includes social sustainability, expressed in personnel policy, which includes management and care for all employees, but especially the key ones, as the functioning of the company is based on them. If the company fails to properly manage and maintain them, these employees may leave the company, which may have significant implications in very sensitive areas such as business strategy, achievement of goals, company culture, or the morale of ordinary employees. The company may suffer significant financial loss by the departure of key employees, its economic (and thus environmental) sustainability or even its existence may be jeopardized. On the other hand, well-managed, strategically linked and well executed management of key employees and positions becomes a significant competitive advantage for successful companies and can contribute to its sustainable business.

We have also replaced the word sustainability with sustainable business in several places throughout the article.

The authors had the paper proof read and the errors have been corrected.

Authors

Reviewer 4 Report

Very interesting paper, well written using academic standards. 

Some minor suggestions:

add study discussion in relations to the previous studies the conclusion is too wide, move some findings table 8 and 9 to previous part. Add some recommendations 

Author Response

Dear reviewer,

thank you very much for the review and your suggestions. Based on your recommendations we moved the table 8 and 9 from Conclusion into the previous part so that the Conclusion is not so long.

At the end of Chapter 5. Practical Implications we have added the following paragraph:  

In addition to the conclusions already mentioned in Chapter 3, the interviews conducted within the focus groups with the representatives of the companies involved in the research study showed that companies were most concerned about retaining key employees. In order to ensure that key employees are motivated to work for the company and want to stay there, the company should strive to be an attractive employer, a high-quality employer, an employer with the reputation of a sustainable business, a place where people like to work. The authors of the article present some practical recommendations on how to achieve this in the company: e.g. to have good leadership, provide key employees with wide opportunities for personal development, as these people want to improve, develop their skills, increase their education; show them respect, appreciation, praise them for their good work; provide empowerment, encouragement and attractive pay and above-standard benefits; plan tailor-made careers for them; Robert Half survey confirms the importance and correctness of the tools for retaining key employees in the company, stating that about 35% of key employees leave because of dissatisfaction with the management, Growth is reported by 33% of key workers and that low recognition or underpayment is the third most common reason for dismissal reported by 13% of key workers [29].

In connection with the paragraph above, we have added another source to References:

Robert Half. It´s time we all work happy. Retrieved from: https://www.indeed.com/cmp/Robert-Half/reviews (accessed on 15 December 2019).

The authors did not add any study discussion in relation to the previous studies, because the best of authors’ knowledge, there are no research studies, which would deal with the rate of usage of the key positions and employees management system in companies neither in the Czech Republic nor abroad as specified in the penultimate paragraph of Part 1. Introduction.

Authors

Round 2

Reviewer 2 Report

Dear authors,

I was very interesting to read your manuscript. I am glad that you have added information about interviews you have done with managers in the manuscript and thus refine the results of your research study.

Reviewer 3 Report

Dear Authors,

the additions you have made to the manuscript fill in the gaps that I pointed out to you, so I now believe that the manuscript is suitable for publication.